Carbon monoxide poisoning: beyond survival - mortality, morbidities, and risk factors, a Turkey sample

Öz Erdoğan 1
Küçükkelepçe Osman osmankkelepce@hotmail.com 2
Kurt Osman 2
Vural Aşkı 3
1 Family Medicine, Adiyaman Provincial Health Directorate , Adıyaman , Turkey
2 Public Health, Adiyaman Provincial Health Directorate , Adıyaman , Turkey
3 Internal Medicine, Adiyaman University , Adiyaman , Turkey
Kibayashi Kazuhiko
Electronic publication date: 2023 Sep 28
Publication date: 2023
Volume: 11
Electronic Location ID: e16093
Received 2023 May 15; Accepted 2023 Aug 23
Copyright: ©2023 Öz et al.
Copyright year: 2023
Copyright holder: Öz et al.
License: This is an open access article distributed under the terms of the Creative Commons Attribution License, which permits unrestricted use, distribution, reproduction and adaptation in any medium and for any purpose provided that it is properly attributed. For attribution, the original author(s), title, publication source (PeerJ) and either DOI or URL of the article must be cited.
License URL: https://creativecommons.org/licenses/by/4.0/

Keywords: Carbon monoxide, Carbon monoxide poisoning, Poisoning, Unintentional poisoning, Toxicity

Funding: The authors received no funding for this work.

==============================
Background

We aimed to investigate the effect of poisoning on mortality leading to new morbidities in people who survived the poisoning.

Methods

The descriptive-retrospective study evaluated all carbon monoxide poisoning cases between 2012 and 2022 in the Adiyaman. For the fatality, all cases were followed up through Turkey’s death notification system until the end of 2022. One-year health records of cases treated as inpatients in Adiyaman hospitals were analyzed for nine diagnoses. A total of 4,395 carbon monoxide cases, recorded over 11 years, were all noted to be accidental cases.

Results

The rate of carbon monoxide poisoning in Adıyaman was calculated as 63.2 per hundred thousand. A total of 87 (2%) of the cases died. The population’s hospitalization rate was 1.71, while the mortality rate was 1.25 in a hundred thousand. Among the cases, the hospitalization rate was 2.7, and the admission to intensive care rate was 1.7. The fatality rate was 6.5% for those hospitalized and 12.2% for those admitted to the intensive care unit. The highest fatality rate was 65.5% in patients aged 65 and above. One out of five morbidities was developed in 8.4% of cases within one year. The fatality rate of those who developed morbidities (40%) was higher than those who did not (5.5%). Being male posed a 1,886-fold risk for mortality, and each increase in age posed a 1,086-fold risk for mortality.

Conclusion

Individuals who had carbon monoxide poisoning should be followed up closely for one year after poisoning due to the possibility of the emergence of new morbidities that increase the risk of mortality.

Introduction

Carbon monoxide (CO) poisoning is one of the most common poisonings. Mattiuzzi & Lippi (2020) found the incidence as 137 per million and the risk of death 4.6 in a worldwide epidemiological evaluation conducted in 2019. Although there was no change in the incidence of CO poisoning between 1992 and 2017, they reported that the risk of death decreased by 36%. Forés & Hamnett (2018) stated that deaths due to CO poisoning are more common in men than women since men are more in contact with CO. Mattiuzzi & Lippi (2020), on the other hand, argued that although there was no difference in gender in terms of incidence, the death rate due to CO poisoning in men was twice that of women.

In Japan, between 2,000–5,000 people die annually from stoves poisoning (Kinoshita et al., 2020). Wood and coal stoves, geysers, and bottled gas are frequently used in Turkey. However, no nationwide record of CO poisoning has been disclosed for more than ten years; there are studies based on the data from the emergency department of one hospital or autopsy data from local forensic medicine institutes (Akkose et al., 2010; Uysal et al., 2013). In a study collecting data sources was controversial; data obtained from a news agency that had been collecting the news from national and local newspapers and TV channels, the risk of death due to carbon monoxide in Turkey was found to be 0.35/100.000 between 2008 and 2017, and Adiyaman was not in the first 15 cities with the highest risk (Can et al., 2019). According to 2010 data, in a study conducted throughout Turkey, Adiyaman was the ninth city in the country with the highest number of cases per population (Metin et al., 2011).

CO, present in trace amounts in our air, is a colorless, tasteless, odorless, non-irritating toxic gas called the “silent killer” (Mattiuzzi & Lippi, 2020). It is found in the smoke formed due to the incomplete combustion of fuels having “carbon” in their contents, such as bottled gas, natural gas, kerosene, and gasoline, especially coal, and wood (Bleecker, 2015).

Since CO cannot be detected during respiration, it causes poisoning much faster than other gases. It is most commonly caused by using items such as a stove for heating or a geyser to obtain hot water in indoor settings with insufficient ventilation (Incekaya et al., 2017). Furthermore, there is a risk of CO poisoning where exhaust fumes are inhaled and in environments with misplaced gasoline-powered generators, iron foundries, ice rinks where propane-powered resurfacing machines are used, and house fires (Bleecker, 2015; Hampson et al., 1994).

Casualties do not realize that they inhale high CO during poisoning. CO poisoning leads to death or permanent defect by causing damage to hypoxia-sensitive organs and tissues such as the brain and heart. Neurological damage may occur two days to 6 weeks after exposure to carbon monoxide (Uysal et al., 2013). Common symptoms include loss of consciousness, dizziness, headache, nausea, and vomiting. Less frequently seen ones are confusion, lethargy, weakness, respiratory distress, chest pain, abdominal pain, ataxia, nervousness, vision disorders, seizures, and incontinence. Ataxia, dementia, Parkinson’s, lack of concentration, and abnormal behaviors are some examples of permanent neurological damage (Eichhorn, Thudium & Jüttner, 2018).

A minimal amount of CO is also produced in the human body. CO, emerging endogenously due to hemoglobin catabolism, leads to less than 1% COHb formation. Endogenous CO is a neurotransmitter and regulator for other neurotransmitters and hormones (Verma et al., 1993).

After CO enters the body with respiration, it has an affinity for hemoglobin 200 times more than oxygen, binds to hemoglobin, and forms carboxyhemoglobin (COHb). COHb leads to tissue hypoxia by reducing the oxygen-carrying capacity and oxygen release to the tissues (Bleecker, 2015; Incekaya et al., 2017). At COHb levels below 5%, the physiological compensation of increased blood flow and oxygen release prevents hypoxia. This blood flow at the beginning continues until loss of consciousness due to hypotension and ischemia in the cerebral arteries. The resulting loss of consciousness might lead to delayed neurological sequelae. Unless the COHb rate in the blood exceeds 20%, there is no significant change in brain oxygenation, while any value above 51% leads to seizures, coma, severe acidosis, and death (Bleecker, 2015; Lo et al., 2007).

Anamnesis is the most valuable finding in CO poisoning since no specific finding may exist in a physical examination. Also, if a long time has passed after exposure to carbon monoxide or supplemental oxygen therapy has been given, the CO level might be misleadingly low (Hampson & Hauff, 2008).

The present study aimed to investigate the effect of carbon monoxide poisoning on mortality after new morbidities in people who survived the poisoning. In order to examine the long-term effects of carbon monoxide poisoning on the person, the development of neurological, cardiological, psychiatric, and endocrinological morbidities after carbon monoxide poisoning was investigated; in this way, the effect of carbon monoxide poisoning on mortality leading to the emergence of new morbidities in people who survived the poisoning was tried to be revealed.

Materials and Methods

This descriptive-retrospective study evaluated all CO poisoning cases that occurred in 11 years between 2012 and 2022 in Adiyaman, with a population of 632,148 in Southeast Turkey.

As a result of the investigation, it was determined that there was a total of 4,395 carbon monoxide poisonings in 11 years in Adiyaman. It is planned to reach the entire universe without conducting a sample calculation that collects Hospital Information Management System (HIMS) data from all eight state hospitals in the province.

The HIMS contains sociodemographic information of individuals who sought medical care at the hospital and health records encompassing examination details, laboratory results, imaging findings, surgeries, vaccinations, treatments, and medication data. Each time an individual visits the hospital, new information is added to their health file, allowing access to their lifelong health records, both for inpatients and outpatients. This information is electronically transmitted to the Ministry of Health’s database daily. As a result, the HIMS is integrated with all provincial hospitals and connected to the Death Notification System, as it regularly sends data to the Ministry of Health. While Provincial Health Directorates can access comprehensive patient data from birth to death within their respective provinces, the Ministry of Health can instantly access health data for all individuals nationwide within the electronic environment.

The study was conducted between 15 December 2022 and 15 January 2023, and the researchers had no access to the names of the cases included in the study.

All patients and fatalities were the ones with the “T58- Toxic effect of carbon monoxide” classification according to the “International Classification of Diseases (ICD)-10 code” in the official records. It was not taken into consideration with which complaint the patients were brought to the hospital by themselves or others—persons not diagnosed with T58 were excluded from the study. It was also not considered whether the clinicians diagnosed CO poisoning by anamnesis or by determining the COHb level.

For the fatality, all cases were followed up through Turkey’s death notification system until the end of 2022. One-year health records from the HIMS were analyzed for cases treated as inpatients in Adiyaman hospitals. Based on previous studies, the analysis focused on nine diagnoses that are likely to occur after carbon monoxide poisoning: cerebrovascular accident, dementia, Parkinson’s, ataxia, depression, anxiety, arrhythmia, myocardial infarction (MI), and hypothyroidism. Suppose any of these nine diseases were diagnosed on a date after CO poisoning. It was accepted that this morbidity occurred after poisoning. These diagnoses were not evaluated as poisoning-related morbidity if present before the poisoning.

Statistical analysis

Data analysis was performed with the SPSS 22 package program. In the study, descriptive data were shown as n and percentile values for categorical data, while they were shown as mean ± standard deviation (mean ± SD) values for the continuous data. Chi-square analysis (Pearson Chi-square) was used to compare categorical variables between groups. The Kolmogorov–Smirnov test evaluated whether the continuous variables had a normal distribution. Mann–Whitney U-test was used to compare paired groups, and the Kruskal-Wallis test was used to compare more than two variables. Logistic regression analysis was used to calculate the mortality risk. Significant ones in the univariate analysis were included in the model for multivariate. The statistical significance level in the analysis was accepted as p < 0.05.

Ethics statement

The study was conducted by the Declaration of Helsinki and approved by the Non-interventional Clinical Research Ethics Committee of Adiyaman University (protocol code 2022/9-1 and date of 13.12.2022). Since this is a retrospective study, obtaining any verbal or written consent from the participants was impossible.

Results

A total of 4,395 carbon monoxide poisoning cases were found to occur in Adiyaman as per the examination of 11 years of data in this study. There was CO poisoning at a rate of 63.2 per hundred thousand in Adiyaman, with an annual average of 399.5 cases. All the cases in the study were accidental, with no cases of suicide or murder. The mean age of the cases was 27.1 ± 20.1 years (min = 1-max = 100). The hospitalization rate per 100,000 people was found to be 1.71. According to the discharge information of 199 patients, 118 (59.3%) were discharged after being cured, 58 (29.1%) with a referral to another institution, 7 (3.5%) were with the treatment, 3 (1.5%) voluntarily, and 13 (6.5%) of them died. Fifty patients among the referred ones were transferred to the hospital in Adiyaman city from the district hospitals, and other 8 (13.8%) were referred to hospitals outside of Adiyaman. 7 of 8 patients were transferred to institutions in other cities without being admitted to any hospital in Adiyaman. The only other case was transferred after he/she spent four days in the hospital’s intensive care unit (ICU) in Adiyaman. The mean hospital stay of the patients was 2.6 ± 2.5 days, and 1.7% of the cases were admitted to the ICU. 87 (2%) of all cases died, of which 70.1% died in the hospital, 25.3% at home, and 4.6% at different locations. The mean time from diagnosis to death was 35.5 ± 30.7 months. When one-year health records of cases treated as inpatients in Adiyaman hospitals were analyzed for nine disease diagnoses, morbidity was detected in 10 (8.4%) of 119 cases. When these morbidities were examined in detail, only five of these nine diseases were diagnosed, which were 3 (2.5%) Cerebrovascular accidents, 3 (2.5%) Parkinson’s disease, 2 (1.7%) anxiety, 1 (0.8% depression), and 1 (0.8%) arrhythmia (Table 1).

Table 1 Characteristics of carbon monoxide poisoning cases.

		Number	%	
Age (years), Mean ± SD	27.1 ± 20.1	
Gender	Female	2,546	57.9	
Male	1,849	42.1	
Area of residence	City	1,106	25.2	
District	1,516	34.5	
Town/Village	1,773	40.3	
Type of treatment	Outpatient	4,276	97.3	
Inpatient	119	2.7	
Type of discharge	Cured	118	59.3	
Referred	58	29.1	
Treated	7	3.5	
Death	13	6.5	
Voluntary	3	1.5	
Duration of hospital stay (days), Mean ± SD	2.6 ± 2.5	
Admitted to the ICU	Yes	74	1.7	
No	4,321	98.3	
ICU stay (days), Mean ± SD	2.3 ± 1.2	
Death	Yes	87	2.0	
No	4,308	98.0	
Place of death	Hospital	61	70.1	
Home	22	25.3	
Other	4	4.6	
Time from diagnosis to death (months), Mean ± SD	35.5 ± 30.7	
Morbidity presence	Yes	10	8.4	
No	109	91.6	
	Cerebrovascular accident	3	2.5	
Parkinson’s disease	3	2.5	
Depression	1	.8	
Anxiety	2	1.7	
Arrhythmia	1	.8	
None	109	91.6	

A total of 87 of the cases died. So, the death rate from carbon monoxide poisoning was calculated as 1.25 per hundred thousand for the population of Adiyaman, with 7.9 deaths annually.

The distribution of cases according to age showed that the highest rate was between 20–39 (30.1%), and the lowest rate was among those aged 65 and above (6.1%). Among the 87 cases with a fatality, the highest fatality rate was observed in the 65 and above age group (Fig. 1).

Figure 1 Distribution of cases and deaths by age (%).

Six (6.9%) of the cases with fatality died at the scene due to carbon monoxide poisoning, while five (6.2%) of the remaining 81 deaths occurred within the first month, eight (9.9%) within the first two months, 12 (14.5%) within the first six months, and 21 (25.9%) within the first year.

When the distribution of the cases was analyzed by month, the highest poisoning rate was in January (34.2%), December (22%), and February (17%), and the lowest in August (0.2%), June (0.4%), and May (0.5%), respectively (Fig. 2). When considering the seasons, the highest rate was observed during winter (73.3%), while the lowest rate was observed during summer (1.4%) (Table 2).

Figure 2 Distribution of cases by month (%).

The mean age of the ones who died was significantly higher than the mean age of the ones who recovered (p < 0.001). The fatality rate of males (2.5%) was significantly higher than that of females (1.6%) (p = 0.039). The fatality rate of inpatients (8.4%) was found to be significantly higher than that of outpatients (1.8%) (p < 0.001). The fatality rate of those who stayed in the ICU (%12.2) was significantly higher than those who did not (1.8%) (p < 0.001). The fatality rate of those with morbidity (40%) was significantly higher than the rate of those without any morbidity (5.5%) according to the five identified diseases (p = 0.004). The highest fatality rate was observed in those with Parkinson’s disease (66.7%), and there was a significant difference between diagnoses in terms of death (p = 0.005) (Table 3).

Table 2 Temporal distribution of CO poisoning patients’ admissions to hospital.

		Number	%	
Hour	00:00:00–07: 59:59	1,795	40.8	
08:00:00–15: 59:59	2,026	46.1	
16:00:00–23: 59:59	574	13.1	
Season	Spring	792	18.0	
Summer	61	1.4	
Autumn	321	7.3	
Winter	3,221	73.3	

Table 3 Comparison of death presence according to various parameters.

Data shows the exitus presence according to various parameters. The p-value is bold if the analysis was significant.

		Death presence	No death	P	
		Number	%	Number	%		
Age (years), Mean ± SD	65.5 ± 22.1	26.4 ± 19.3	<0.001 ∗	
Gender	Female	41	1.6	2,505	98.4	0.039 ∗∗	
Male	46	2.5	1,803	97.5	
Area of residence	City	25	2.3	1,081	97.7	0.121∗∗	
District	21	1.4	1,495	98.6	
Town/Village	41	2.3	1,732	97.7	
Type of treatment	Outpatient	77	1.8	4,199	98.2	<0.001 ∗∗	
Inpatient	10	8.4	109	91.6	
Type of discharge	Cured	3	2.5	115	97.5	0.072∗∗	
Referred	6	10.3	52	89.7	
Treated	1	14.3	6	85.7	
Voluntarily	0	0	3	100.0	
Staying in the ICU	Yes	9	12.2	65	87.8	<0.001 ∗∗	
No	78	1.8	4243	98.2	
Morbidity presence	Yes	4	40.0	6	60.0	0.004 ∗∗	
No	6	5.5	103	94.5	
Diagnoses received within one year after hospitalization	Cerebrovascular accident	1	33.3	2	66.7	0.005 ∗∗	
Parkinson’s Disease	2	66.7	1	33.3	
Depression	0	.0	1	100.0	
Anxiety	1	50.0	1	50.0	
Arrhythmia	0	0	1	100.0	
None	6	5.5	103	94.5		
Duration of hospital stay (days), Mean ± SD	3.0 ± 1.7	2.6 ± 2,5	0.256∗	
Duration of ICU stay (days), Mean ± SD	2.3 ± 1.0	2.3 ± 1.2	0.685∗	

The time from diagnosis to death in outpatients was significantly higher than inpatients (p = 0.009). It was significantly lower in those staying in the intensive care unit than in those who did not (p = 0.026) (Table 4).

Table 4 Comparison of the time from diagnosis to death according to various parameters.

Data shows the time from diagnosis to death according to various parameters. The p-value is bold if the analysis was significant. An asterisk (*) indicates Mann Whitney U test; two asterisks (**) indicate Kruskal Wallis test.

		Time from diagnosis to death (months)	P	
		Mean ± SD		
Gender	Female	36.3 ± 34.6	0.862∗	
Male	34.7 ± 27.1	
Area of residence	City	32.2 ± 30.0	0.068∗∗	
District	47.9 ± 29.4	
Town/Village	31.1 ± 30.8	
Type of treatment	Outpatient	37.8 ± 30.3	0.009 ∗	
Inpatient	17.2 ± 29.2	
Type of discharge	Cured	30.7 ± 30.6	0.149∗∗	
Referred	6.0 ± 7.9	
Treated	15.0 ±-	
Staying in the ICU	Yes	19.0 ± 30.4	0.026 ∗	
No	37.4 ± 30.4	
Place of death	Hospital	34.0 ± 30.3	0.468∗∗	
Home	36.1 ± 30.9	
Other	54.8 ± 38.0	

According to the univariate analysis in the logistic regression analysis to calculate the mortality risk, being male increases the risk of mortality 1.559 fold (95% CI [1.019–2.385]), one year increase of age increases it 1.087 fold (95% CI [1.074–1.101]), receiving inpatient treatment increases it 5.003 fold (95% CI [2.520–9.932]) and staying in intensive care increases it 7.532 fold (95% CI [3.622–15.663]). The ones found to be significant in the univariate analysis were used in the multivariate analysis, and being male posed a 1,886-fold (95% CI [1.187–2.996]) risk for mortality. Each age increase had a 1,086-fold (95% CI [1.072–1.099]) risk for it (Table 5).

Table 5 Logistic regression analysis of the presence of mortality.

The p-value is bold if the analysis was significant.

	Univariate	Multivariate	
	B	p	OR	%95 CI	B	p	OR	%95 CI	
Gender (ref = female)	444	0.041	1.559	1.019–2.385	634	0.007	1.886	1.187–2.996	
Age	084	<0.001	1.087	1.074–1.101	082	0.001	1.086	1.072–1.099	
Type of treatment (ref=outpatient)	1.610	<0.001	5.003	2.520–9.932	138	0.897	1.148	141-9.338	
ICU (ref=non-staying)	2.019	<0.001	7.532	3.622–15.663	586	0.607	1.797	193–16.754	

Discussion

The rate of intentional CO poisoning cases is rare in Turkey. Only 0.3% of cases were found to be intentional in one study (Akkose et al., 2010). There were no intentional cases of suicide or murder in the present study as per the examination of all the cases included, which may be due to intentional cases with a fatality being transferred to the Forensic Medicine Institute for autopsy directly without being taken to the hospital, or the fact that deaths cannot be classified as CO poisoning, since the cause of death cannot be determined without an autopsy.

In the present study, the CO poisoning rate was found to be 63.2 per hundred thousand, much higher than the data in the literature. In an epidemiological study conducted worldwide in 2019, it was found to be 13.7 per hundred thousand (Mattiuzzi & Lippi, 2020). The rate was 14 per hundred thousand in 10,154 cases in Turkey in 2010 and 4.3 per hundred thousand for accidental cases in Ankara and its surrounding 15 cities between 2001–2011 (Metin et al., 2011). In a study conducted in Iran, the rate was reported as 38.91 per hundred thousand (Roca-Barceló et al., 2020). In some of the villages and districts with lower economic income in Turkey, there is no natural gas. Even if it is available, the only way of heating is traditional wood and coal stoves in most houses. Geysers working with gas bottles are used to heat water in the bathrooms.

Therefore, the incidence of CO poisoning in rural areas was higher than the urban areas. Most of the CO poisoning cases (75%) in the present study were found to reside in rural areas outside the urban areas, supporting the study of Uysal et al. (2013). Adiyaman is in the Southeast region of Turkey, which has the lowest socioeconomic level. Although natural gas usage rates are increasing daily, wood and coal stoves are still used frequently, especially in rural areas. In the study of Metin et al. (2011) in 2010, the highest number of CO poisoning cases according to the population occurred in Kilis, also a southeastern city, with 199.80 cases per hundred thousand. According to the population, the rate in Adiyaman was 36.60 cases per hundred thousand. Although Adiyaman was the 11th city with the highest number of cases, with 216 cases, and the ninth city according to the population, we can not fully explain the incidence of CO poisoning in Turkey was only 14 cases per hundred thousand. Many factors, such as climate, socioeconomic, and education levels among 81 cities, might be the reasons for significant differences in incidence between the cities.

Furthermore, Metin et al. (2011) reported 10,154 CO-poisoning cases in 2010. Even though the rate of natural gas use was much lower, they noted no deaths in eight of the 20 cities with the highest cases. Adiyaman was one of these eight cities. In this study, the annual mean number of fatalities was 7.9. Metin et al. (2011) reported only 39 deaths in 12 20 cities with the highest cases. Besides the authors’ statement of this particular study that they had suspicions of a deficiency in the number of deaths, we think there might have been a deficiency in the number of cases and deaths in the study. Accordingly, CO poisoning in Turkey in 2010 should have been higher. However, in some years, the winter season is colder or warmer, and the number of days when the wind is intense changes. These changes might have an impact on the number of CO poisonings. The 11-year data in the present study, different from the studies with 1-year data, eliminates the possibility of a random cold or hot year.

In the present study, 73.2% of the poisonings occurred in winter and 7.4% in autumn. A study conducted in Bursa, located northwest of Turkey, showed that 64.6% of the cases occurred in the winter season (Akkose et al., 2010). A study in France determined that 84% of poisonings occurred in the autumn-winter seasons (between October and March) (Sam-Laï, Saviuc & Danel, 2003).

In this study, the least poisoning rate was 6.1% for those aged 65 or above. However, this group had the highest fatality rate, and the mean age of cases was 27.1 years. In their epidemiological study, Mattiuzzi & Lippi (2020) determined that the incidence of CO poisoning had two definite peaks between the ages of 0–14 (31% of all cases) and those between the ages of 20–39 (34% of all cases). Both groups had the highest incidence of poisoning in the present study, with 34.9% between 0–14 years old and 30.1% between 20–39 years old. Similar to other cities in the Southeastern region of Turkey, Adiyaman, with a high fertility rate, has a high young population. The high rate of young population in the city might be why most cases of CO poisoning occur in the younger population.

The hospitalization rate for accidental CO poisoning cases between 2001 and 2010 for the entire population in England was 0.49/100,000. In addition, the incidence of cases and hospitalization rates in rural areas was higher than in urban areas (Liao et al., 2019). In this study, the hospitalization rate per population was higher (1.71/100,000) than in England. However, in the study covering the cases between 2002 and 2016, the mean number of annual accidental CO poisonings in England was 107.8, whereas this number was 399.5 in Adiyaman. The higher hospitalization rate in Adiyaman compared to England might be related to the number of cases rather than the severity or clinician preference.

However, the present study’s data covers all the hospitals in Adiyaman, including the small-scale hospitals in its districts. The hospitalization status of the seven patients transferred directly to other cities without being hospitalized in Adiyaman could not be determined. If the data could be accessed, patients who were transferred to hospitals outside the city with better facilities would increase the rate of hospitalization and intensive care because of their critical situation. However, none of those eight patients transferred out of the city died. Regarding nine diseases identified, one of the ten patients diagnosed (anxiety) in Adiyaman hospitals was one of the patients transferred out of the city.

In the present study, 87 of 4,395 CO poisoning cases in 11 years died, and the mortality rate in Adiyaman was 1.25 per hundred thousand. In a study worldwide, the risk of death from CO poisoning was found to be 0.46 per hundred thousand, while it was 0.35 per hundred thousand in a study conducted across Turkey (Mattiuzzi & Lippi, 2020; Can et al., 2019).

According to the data from the Forensic Medicine Institute in Ankara, a total of 380 people, 203 in Ankara and 177 in 15 cities reporting to this institute, died between 2001–2011 due to carbon monoxide poisoning (Uysal et al., 2013). In the present study, the fatality rate was 6.5% in hospital admissions and 12.2% in ICU patients. In their study published in 2010, Akkose et al. (2010) reported that the fatality rate due to carbon monoxide poisoning in patients admitted to the hospital ranged from 2.6% to 9.8% in the literature. However, in Iran, Nazari, Dianat & Stedmon (2010) reported in their study, also published in 2010, that 346 people died of 3,078 CO poisonings in Northwest Iran between 2003 and 2008, so the fatality rate was 11.24%. In a study conducted in China, the fatality rate was 2.5% in hospital admissions and 14.3% in those admitted to the ICUs (Huang et al., 2014). In Turkey, the fatality rate was 3.3 in 305 patients who visited the emergency department of a hospital in Bursa (Akkose et al., 2010). This rate was 3–5% in France and 9.5% in Iran (Sam-Laï, Saviuc & Danel, 2003; Roca-Barceló et al., 2020).

Hampson & Weaver (2007) stated that deaths from CO poisoning were more common in older adults. In their study, Mattiuzzi & Lippi (2020) argued that the risk of death from carbon monoxide poisoning was high in older adults and infants, and the risk increased with age. In the present study, the highest death rate was seen in people aged 65 or above, with 65.5%. However, unlike Mattiuzzi & Lippi’s finding, only 1.1% of deaths were children aged between 0 and 5 years, despite accounting for 10.7% of all the cases in the present study. The mean age of the ones who died was significantly higher than the mean age of the ones who survived. In the multivariate analysis of age, gender, treatment method, and ICU admission criteria, each age increase had a 1,086-fold risk for mortality.

The effects of chronic neurological damage begin to appear after asymptomatic days or weeks following exposure to CO due to the aggravation of existing chronic diseases or new morbidities triggered by the poisoning; the risk of death becomes high, especially in the first month, within 1–6 months, and 6–12 months afterward. In the present study, after carbon monoxide poisoning, six (6.9%) of the cases died at the scene, while five (6.2%) of the remaining 81 deaths occurred within the first month, eight (9.9%) in the first two months, and 12 (14.5%) in the first six months. The mean time from the diagnosis of poisoning to death was three years (35.5 ± 30.7 months). In the study of Huang et al. (2014) 1.93% of the patients followed up between 1999 and 2010 died during the follow-up, and the death rate was 5.24 times higher than the control group (Rose et al., 2017). The complaint-free period after poisoning leads to false reporting of these sequelae and deaths caused by these sequelae with another diagnosis unrelated to carbon monoxide poisoning.

Eichhorn, Thudium & Jüttner (2018) stated that delayed neurological damage due to CO poisoning reveals itself in the form of ataxia, dementia, Parkinson’s disease, lack of concentration, and abnormal behaviors. Delayed neurological damage is between 15% and 40% (Han et al., 2021). Increased risk of neuropsychiatric sequelae, arrhythmia, MI, coronary artery disease, heart failure, hypothyroidism, diabetes mellitus, and hypertension was detected in the follow-up period after CO poisoning (Lee et al., 2015; Zavorsky et al., 2014). In the present study, the health data of 119 hospitalized patients in Adiyaman hospitals were scanned for one year for cerebrovascular accident, dementia, Parkinson’s disease, ataxia, depression, anxiety, MI, arrhythmia and hypothyroidism, and at least one of these diseases was detected in 10 (8.4%) patients. Three (2.5%) patients were diagnosed with cerebrovascular accident (CVA), three (2.5%) with Parkinson’s disease, one (0.8%) with depression, two (1.7%) with anxiety, and one (0.8%) with arrhythmia. The fatality rate of those with morbidity (40%) was significantly higher than those without (5.5%). The highest number of deaths occurred in those with Parkinson’s disease, with 66.7%. The mechanisms by which CO poisoning paves the way for developing diseases that cause mortality should be clarified by further studies.

In the present study, the fatality rate of males (2.5%) was significantly higher than that of females (1.6%), and according to multivariate analysis, being male posed a 1,886-fold risk for mortality. In one study, the mortality due to carbon monoxide poisoning in the general population was twice as high in men as in women. This difference was attributed to the shorter COHb elimination times in women than in men. Because of the short elimination period, women could tolerate carbon monoxide better (Zavorsky et al., 2014). We think it would be helpful to examine the effect of gender on fatality in CO poisoning with other studies to be conducted in the future.

Stove-related poisoning occurs, especially at night. In cases where the piping setting of the stove, especially in windy weather, cannot provide good ventilation, CO gas leaks into the room from the incompletely burned wood and coal. Households who do not realize that the stove has been fully lit or not at night while asleep are exposed to the harmful effects of carbon monoxide. We did not find any study in the literature comparing the time of the day of their admission to the hospital due to CO poisoning. In the present study, 40.8% of the cases were admitted to the hospital between 00–08 h and 46.1% between 08–18 h. Surprisingly, 40.8% of them visited the hospital before 8 am. Individuals who get up early for morning prayers or things to do in their villages may have survived due to not being further exposed to Carbon monoxide and have ensured that their household members were admitted to the hospital with them.

In the present study, the morbidities of 119 hospitalized patients were examined for one year to diagnose nine diseases obtained from Adiyaman Hospitals. The morbidity follow-up is based on a small sub-sample and led to the identification of ten cases only; therefore, results on the morbidity follow-up may need to be reliable. Additionally, the hospitalization of seven severe cases in service and intensive care units outside the province, who were transferred without being admitted to any hospitals in Adiyaman, could not be evaluated. Considering this information, hospitalization and morbidity rates may be higher for the 4,395 cases.

Conclusions

Although almost all CO poisoning can be prevented with simple precautions, it is still a severe cause of death in regions of Turkey where traditional heating and water heating methods are used. The present study may have shown that CO poisoning increases the possibility of new morbidities (8.4% of cases) with an unclear mechanism that increases mortality risk. Since morbidities such as cerebrovascular accidents, ataxia, dementia, Parkinson’s, depression, anxiety, and arrhythmia leading to death might occur weeks or months after poisoning, people diagnosed with CO poisoning should be followed up closely for about one year. Furthermore, clinicians should consider CO poisoning in the etiology of sudden neurologic, cardiologic, psychiatric, and endocrinologic disorders. In addition, the present study showed that women and the elderly have a greater risk of mortality from CO poisoning, and 29.9% of deaths occurred outside the hospitals, indicating that CO poisoning is a multidisciplinary health problem that concerns many institutions.

Supplemental Information

Supplemental Information 1 Raw data

Click here for additional data file.

We acknowledge the support given by the IT staff of Adiyaman Provincial Health Directorate İsmail Turhan and Yaşar Dikici during the data collection process.

Additional Information and Declarations

Competing Interests

Author Contributions

Human Ethics

Data Availability

The authors declare there are no competing interests.

Erdoğan Öz conceived and designed the experiments, prepared figures and/or tables, authored or reviewed drafts of the article, and approved the final draft.

Osman Küçükkelepçe conceived and designed the experiments, prepared figures and/or tables, authored or reviewed drafts of the article, and approved the final draft.

Osman Kurt analyzed the data, prepared figures and/or tables, and approved the final draft.

Aşkı Vural conceived and designed the experiments, analyzed the data, prepared figures and/or tables, and approved the final draft.

The following information was supplied relating to ethical approvals (i.e., approving body and any reference numbers):

The Non-interventional Clinical Research Ethics Committee of Adıyaman University approved the study with the protocol code 2022/9-1 and date of 13.12.2022.

The following information was supplied regarding data availability:

The raw data is available in the Supplemental File.

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
