# Peer review of "Carbon monoxide poisoning: beyond survival - mortality, morbidities, and risk factors, a Turkey sample"

_PeerJ, doi:10.7717/peerj.16093_

## Round 0.1 · original submission · Major Revisions

Dear Authors:
This is an important study for the care of patients with CO poisoning. I recommend revise the manuscript according to the comments from reviewers. Also consider correcting the following:
1. The term "exitus" in Table 3 is not a commonly used term and is unintelligible.
2. In Table 3 and 4, some rows are hidden and not visible.
3. Please check the low resolution of Figure.
Kazuhiko Kibayashi

·

Basic reporting

All the points for revision are mentioned in the comments box in pdf (please refer PDF)

Experimental design

No comments

Validity of the findings

All the points for revision are mentioned in the comments box in pdf (please refer PDF)

Additional comments

All the points for revision are mentioned in the comments box in pdf (please refer PDF)

·

Basic reporting

The paper aims to address a relevant research question, is well structured and mostly written in clear English. However, a language check is recommended as some sentences are hard to understand. e.g. line 84-85 “However, no record of CO poisoning has been nationwide for more than ten years”.

There is a great redundancy in the text and the tables/figure. The figure bears no legend and - given the redundancy - is not relevant.

Experimental design

The biggest problem with the paper is the study design. The authors seem to make use of 3 data sources: The Hospital Information Management System, the Death Notification System and one-year health records of cases treated as inpatients in hospitals. No information is given how the data were linked and how complete the data could be merged. Whereas the inpatients were selected by specific diagnoses, no such selection seems to have taken place for the mortality data. However, no control group of patients without CO poisoning or of the general regional population was built so there is no way to relate the (later) fatalities to the intoxication nor to compare the risk of mortality/morbidity. Unfortunately, this approach does not allow to reach the aims of the study which is given in the introduction as “The present study investigated both the frequency and short and long-term effects of carbon monoxide poisoning in Adiyaman, a southeastern province of Turkey” and in the abstract as “We aimed to evaluate Carbon monoxide poisonings … especially to investigate the effect of poisoning on mortality leading to new morbidities in people who survived the poisoning.” Besides these aims differ, both are not captured by the title of the paper which furthermore does not account for the regional approach of the study.

Validity of the findings

Methods:
Line 103-105: “In the power analysis made concerning the study of …, n = [DEFF*Np(1-p)]/ [d2/Z21-a/2*N-1)+p*(1-p)] It was found that at least 262 patients should be reached at 95% confidence interval.” This is not understandable as no explanation of the formula is given and the idea why to make power calculation when studying the complete population is not elaborated on. This applies also to the statistical tests used. Why is this considered necessary? Is the study population considered as a sample? Of what?
More information on the study population would be welcomed. How is the HIMS working? Do alle hospitals of the province comply? What information/variables were taken from the HIMS.
How was the link to the Turkeys Death Notification System made? Which diagnosis was taken?
How was the link to the health records of inpatients made? All hospitals included? How was the coverage? How were the diagnoses selected?

Results:
For the calculation of the poisoning rate, it was assumed that the population of the province was constant over 11 years. Is this reasonable?
Line 149: There were more patients with discharge information than inpatients. I would assume that all outpatients would be cured, so which is the population with discharge information and what does the type of discharge tells us e.g. is there “cured” type without treatment? Or type death n=13 in relation to exitus n=87?
119 cases were followed in one-year health records. Is this accidentally the same number than the inpatients of table 1 or is this the same population?
Line 165-167: “A total of 87 of the cases died. So, the death rate from carbon monoxide poisoning was calculated as 1.25 per hundred thousand for … “. Again, I do not understand how these cases are related to the death type of discharge. When cases were from the death notification system how were they identified as result of CO intoxication?
Line 172-176 “Six (6.9%) of the cases with fatality died at the scene due to carbon monoxide poisoning, while 5 (6.2%) of the remaining 81 deaths occurred within the first month, 8 (9.9%) within the first two months, 12 (14.5%) within the first six months, and 21 (25.9%) within the first year. However, different reasons other than CO poisoning were recorded as the causes of death for these eight cases.“ See questions above. Which are “these eight cases”
Table 3 shows 3 cases with discharge type “death” but with “no exitus”.
Paragraph needs more information on the use of data sources and how the populations are linked. It should be better structured according to the research question and data sources.

Discussion: The discussion is rather patchy and is more on the description of other study results but on the discussion of possible reasons for agreement or disagreement. No limitations of the study are mentioned.

Conclusion: “The present study showed that 11.8% of patients discharged from the hospital died within one year because CO poisoning increases the possibility of new diseases (8.4% of cases) with an unclear mechanism that increases mortality risk”. This is rather speculative given the restriction of the study design.

·

Basic reporting

Clear and unambiguous, professional English used throughout.
OK.
Literature references, sufficient field background/context provided.
OK
Professional article structure, figures, tables. Raw data shared.
OK
Self-contained with relevant results to hypotheses.
OK

Experimental design

Original primary research within the aims and scope of the journal.
Research question well defined, relevant & meaningful. It is stated how research fills an identified knowledge gap.
The submission clearly define the research question, relevant and meaningful. The knowledge gap has been identified, and statements made as to how the study contributes to filling that gap.
Rigorous investigation performed to a high technical & ethical standard.
The investigation have been conducted rigorously and to a high technical standard. The research have been conducted in conformity with the prevailing ethical standards in the field.
Methods described with sufficient detail & information to replicate.
Methods described with sufficient information to be reproducible by another investigator.

Validity of the findings

Impact and novelty assessed. Rationale & benefit to literature is clearly stated.
.
All underlying data have been provided; they are robust, statistically sound, & controlled.

Conclusions are well stated, linked to original research question & limited to supporting results.

Additional comments

Any other comment

---

## Round 0.2 · Major Revisions

Dear authors:
1. In abstract and text, the phrase "new diseases" is difficult to understand: is it the onset of another disease not related to CO poisoning? Please clarify.
2. Please revise your manuscript to address the reviewer's comments. All authors should carefully review the manuscript.
Kazuhiko Kibayashi

·

Basic reporting

Dear authors,
Thank you very much for your reply to my questions on the manuscript and for the respective changes in this revision.
The readability for an international audience has improved by this revision. Still, there are some hard-to-understand sentences and typos. I still find the discussion difficult to follow as it seems to jump back and forth a bit. As a reader I would prefer a short summary of the results of this paper first which then will be comprehensively discussed in the light of other studies.

The study aim is given as (line 92-93): “The present study aimed to investigate the effect of carbon monoxide poisoning on mortality leading to new morbidities in people who survived the poisoning.” Should it not read something like: “The present study aimed to investigate the effect of carbon monoxide poisoning on mortality after new morbidities in people who survived the poisoning.”

Line 118: “(%12.2)“

Table 3: N is missing for “City”

Table 5: legend missing e.g. what is GA? Unplausible confidence interval for “type of treatment” and “ICU” e.g. 141-9.338

line 307-308: “Furthermore, 21 (25.9%) died within one year. However, the causes of death were recorded for reasons other than CO poisoning.” According to the rebuttal letter this is wrong and should have been deleted.

The figures bear no information on what is displayed (%?)

Experimental design

The experimental design has been described much clearer now. I understand that a mortality follow-up for a full cohort of CO intoxicated patients (n=4,395) between 2012-2022 was carried out. Additionally, for a sub-cohort (n=119), those marked as “inpatient”, a one-year morbidity follow-up for selected diagnoses (individually after hospital treatment?) took place.

Based on this excellent data base, the study has the potential to give reliable information with respect to the distribution of poisoning cases and the mortality. However, the morbidity follow-up is based on a small sub-sample and led to the identification of n=10 cases only. As no control group design was used there is unfortunately little evidence that these cases show different morbidity than patients without a prior CO intoxication. Results on the morbidity follow-up may therefore be not very reliable.

The specifics of the study are not well captured by the title “Carbon monoxide poisoning: are hospital discharge rates high enough?”. First, hospital discharge rates were rarely used in the study. There seems to be a sub-cohort “Type of discharge” in table 1, different from the above-mentioned cohorts, but information on the kind of this group and how it differs are missing. Second, it is unclear what “high enough” refers to. As a cohort of patients is studied, the follow-up cannot lead to higher rates. Third, it is meant to suggest that the morbidity follow-up gives additional information on the severeness of CO intoxication. However, this is not a problem of a hospital discharge rate and besides highlights the weakest part of the study, given the experimental design.

My recommendation is therefore to stronger link the title and the study aim to the main issue of the paper. I still think that the regionality of the data should also be mentioned in the title as CO intoxication obviously has very different regional characteristics.

Side note: Why is this called a “descriptive-retrospective study” (line 100) in contrast to the use of statistical tests and follow-up data?

Validity of the findings

Line 174 : “The lowest rate was in the cases of 65 or above (6.1%). In the distribution of 87 cases with a fatality, the highest fatality rate was found in the 65 or above group (Figure 1).“ The sentences sound contradictory and misleading as fatality rates do not seem to be shown at all. My impression is that rates and distribution (%) are used synonymous in the paper what can be confusing.

With respect to the conclusions “The present study may have shown that 11.8% of patients discharged from the hospital died within one year because CO poisoning increases the possibility of new diseases (8.4% of cases) with an unclear mechanism that increases mortality risk”, I failed to see where the figure 11.8% comes from. Line 307 reads “Furthermore, 21 (25.9%) died within one year”. In case it refers to the morbidity follow-up this should be mentioned as 98% of the poisoning cases were not included. Basically of those patients with follow-up, 4 fatal cases were affected by the selected diseases against 6 fatal cases without.

---

## Round 0.3 · accepted · Accept

Dear authors:

I have confirmed that the authors have revised the points raised by the reviewers. The manuscript is ready for publication.

Your paper will contribute to the improvement of medical care and life-saving rates for patients with acute carbon monoxide poisoning. I look forward to your continued research.

Kazuhiko Kibayashi